# Analysis of the Effectiveness of a Freight Transport Vehicle at High Speed in a Vacuum Tube (Hyperloop Transport System)

David S. Pellicer *  and Emilio Larrodé

Department of Mechanical Engineering, University of Zaragoza, 50018 Zaragoza, Spain; elarrode@unizar.es
* Correspondence: dasapezu@unizar.es

**Abstract:** This paper shows the development of a numerical analysis model, which enables the calculation of the cargo transport capacity of a vehicle that circulates through a vacuum tube at high speed, whose effectiveness in transport is analyzed. The simulated transportation system is based on vehicles moving in vacuum tubes at high speed, a concept commonly known as Hyperloop, but assuming the vehicles for cargo containers. For the specific vehicle proposed, which does not include a compressor and levitates on magnets, the system formed by the vehicle and the vacuum tube has been conceptually developed, establishing the corresponding mathematical relationships that define its behavior. To properly model the performance of this transport system, it has been necessary to establish the relationships between the design variables and the associated constraints, such as the Kantrowitz limit, aerodynamics, transport, energy consumption, etc. Once the model was built and validated, it was used to analyze the effects of the variation of the number of containers, the operating speed and the tube length, considering the total and specific consumption of energy. After finding the most efficient configuration regarding energy consumption and transport effectiveness, the complete system was calculated. The results obtained constitute a first approximation for the predesign of this transport system and the built model allows different alternatives to be compared according to the design variables.

**Keywords:** mathematical modeling; high-speed transport; freight transport; sustainable transport





## 1. Introduction

The objective of this work is to develop a calculation model that could find the best configuration of a vehicle that transports heavy goods at high speed in a vacuum tube, and thus, obtain greater energy efficiency as well as greater effectiveness in the operation of transport. The process consists of defining a case study under behavioral hypotheses, parameterizing the problem through the behavioral equations corresponding to each of the physical phenomena that occur, and applying the analysis to a predesign of a vehicle that simulates the operation in real conditions. Once the behavior relations of the system are established, the energy consumption, the performance of the system, and the verification of the Kantrowitz limit, are determined. This allows selection of the optimum amount of load to be transported, the most suitable operating speed, and the most appropriate tube length. Once the optimal values of these variables have been obtained, the rest of the main characteristics of the vehicle are determined.

Regarding the vehicle, the infrastructure and their interaction, two options for levitation have been considered: air bearings and electrodynamic suspension (EDS). This work focuses on the latter. The vehicle does not include a compressor to overcome the Kantrowitz limit at near-sonic speeds or airfoils [1,2]; but includes batteries in the rear of the vehicle for the control, and in the EDS rotor. The vehicle also has a mechanical brake for immediate braking in the event of an emergency. Since the usage of standardized containers avoids breaking the cargo down at the terminals (breakbulk shipping), only cargo contained in 20-foot aluminum Dry Van containers is considered and each container must be placed

within a single capsule/pod As for the infrastructure, a linear geometry has been studied, with a straight tube with zero slopes, and which can be 500, 750 or 1000 km long between origin and destination.

Due to work limitations, other issues such as technical and economic feasibility [3], control loops, stability, infrastructure, vehicle structure, heat transfer, EDS geometry and electrical systems are out of scope and may be eligible for additional work.

This research paper can be compared to other cutting-edge work on high-speed transportation systems but differs in some significant respects. These differences are listed below. Reference [4] models a high-speed transportation system and optimizes it, but the system is for passengers instead of cargo and the number of capsules per vehicle is not varied when the energy consumption is minimized. Reference [5] models the system, but it does not use formulae to optimize the system, as the article is only a technology review. Reference [6] models the system and optimizes it, although the system optimization only focuses on levitation and propulsion (electromechanics) and leaves out the other subsystems. References [7–9] also model the system, but their model is used to study dynamics, so the system is not optimized, and capacity is not discussed. Reference [10] discusses transport capacity after building a model, which is based upon logistics relations rather than the physical relations coming from the physical phenomena that occur in the system. Reference [11] focuses on the aerodynamics and thermodynamics of the Hyperloop, but these models also leave out the other subsystems. References [12,13] limit their analyses to aerodynamics, with a different approach (experimental and numerical, respectively), although the other subsystems are also out of scope.

The main contribution of this work is the determination of the most suitable masses and volumes for freight transport using containers in a vehicle that travels at high speed in a vacuum tube levitating on magnets, with the objective of achieving the highest effectiveness possible and allowed. It is important to remark that effectiveness is the sum of efficiency (lower energy consumption per ton and kilometer transported) and efficacy (higher cargo throughput in the transportation system). In order to attain this objective, the procedure of analysis has been altered in regard to the previous research.

Differently to the previous research, the procedure of analysis developed and presented in this article takes into account all of the physical conditions of the problem, adding the restrictions and limitations of the case to be studied. The result is the variation of the parameters sought. In this case, for example, the optimal weight and volume, which allows finding the most appropriate alternative to the proposed criterion, is aimed at the minimum energy consumption. The variation of parameters sought is not very dissimilar to that proposed in references [14,15], which propose the usage of a Monte Carlo method to vary the parameters stochastically, with the main difference being that the variation performed here does not rely on statistics, but on predefined value ranges. In addition, the variation carried out is not automatically refined since the optimization is discussed step by step.

Once the analysis procedure has been validated, the methodology is open to the addition of more restrictions and limitations for future research work.

To conclude with the introduction, it is worth remarking that in order to carry out this research work it has been necessary to review the behavioral theories of the different physical phenomena involved and the extraction of the corresponding behavioral laws. Likewise, it has been necessary to review the most recent research related to the concept of high-speed transport with vehicles in vacuum tubes. These previous studies provide the necessary equations to define the model of a high-speed transport system. This system is made up of three main subsystems: aerodynamics, electromechanics and thermodynamics. The equations that define the relations and limitations of aerodynamics have been extracted from [16–18]. The equations for the electromechanical behavior have been taken from [19–21]. The equations related to the thermodynamic phenomena that occur in the system come from [22,23].

## 2. Materials and Methods

This work follows a deductive method: through the construction of the physical problem to be solved, the behavioral equations of thermodynamics, electromechanics and aerodynamics are applied to the specific case proposed. By establishing the determined limits, the comparison variables that allow an analysis based on the variation of parameters are obtained, and it is this variation of parameters which enables the acquisition of an optimal design. First, the problem to be solved is defined, consisting of the establishment of the behavior laws of a vehicle levitating on magnets in a vacuum tube to be transported at high speed, for which a series of hypotheses have been proposed. These hypotheses are fundamental to delimit the model of the high-speed transport system, which is defined by the physical equations of its main subsystems: aerodynamics, electromechanics and thermodynamics. Second, these equations are interrelated by auxiliary equations that are introduced later, building a system of equations that is solved by mathematical equation solving software. This software allows solution of the system of equations after configuring the input data. The parameters can be varied in the case study: the calculation is carried out by varying one parameter at a time.

### 2.1. Hypthoteses

The following hypotheses have been regarded:

1.  Subsonic speed.
2.  Ideal gas theory, since the compressibility factor is around 1 under the system working conditions.
3.  Isentropic compression as the vehicle moves and the air is compelled to flow into the annulus.
4.  The boundary layer does not separate from the vehicle.
5.  Both acceleration and deceleration are held constant.
6.  The diameter needed to accommodate the load is equal to the diameter of the circumference surrounding a container.
7.  The frontal area of the EDS magnets is negligible with respect to the annulus area.
8.  Active power losses in the EDS are modeled with a single stator resistance.
9.  Any lateral forces generated by the propulsion part of the EDS are not considered. These are inherently stabilizing and low with respect to the propulsion force [17].
10. The average power dissipated by the EDS drag is considered as one third of the maximum during acceleration and braking. This is because the power dissipated first increases and then decreases with speed [20]. If it were linear with speed, then the average power would be half of the maximum, but in this case, it is less, due to this decrease.

### 2.2. Calculation Process

An algorithm consisting of three parallel branches that conflate at a point has been constructed:

*   In the left branch, the power dissipated by aerodynamic drag is computed. For that, the speed of the vehicle and its thermodynamic data are entered. At that given speed, the tube diameter is calculated so that the Kantrowitz limit is prevented. According to the blockage ratio, the power dissipated by aerodynamic drag is computed.
*   In the middle branch, the onboard batteries which feed the rotor of the linear motor are dimensioned. Their dimensioning comes from evaluating their energy density and their discharge time, which depends on the total travel time. In turn, the total travel time relies on the operating speed, acceleration and deceleration of the vehicle through kinematic relations.
*   In the right branch, the power needed to propel and lift the vehicle is calculated. This calculus relies highly on the number of containers (which equals the number of capsules in the vehicle) and their individual masses, which depend on the filling factor

of each container. These data determine how much mass is lifted and propelled and, thus, the power needed for that.

These branches conflate, in order to determine the energy consumption of the vehicle. In this way, the energy consumption is linked to the mass transported and to the operating speed, which allows the finding of relations between the mass flow and the energy needed to maintain that mass flow (always considering that only one vehicle, the one that is to be optimized, is using the tube).

The algorithm is shown in the Figure 1, which shows how the different equation blocks are interrelated. Equation blocks referring to the main subsystems (aerodynamics, electromechanics and thermodynamics) are represented with a bolded contour, while auxiliary equation blocks are represented with a normal contour. The final block is represented with a doubly-bolded contour:

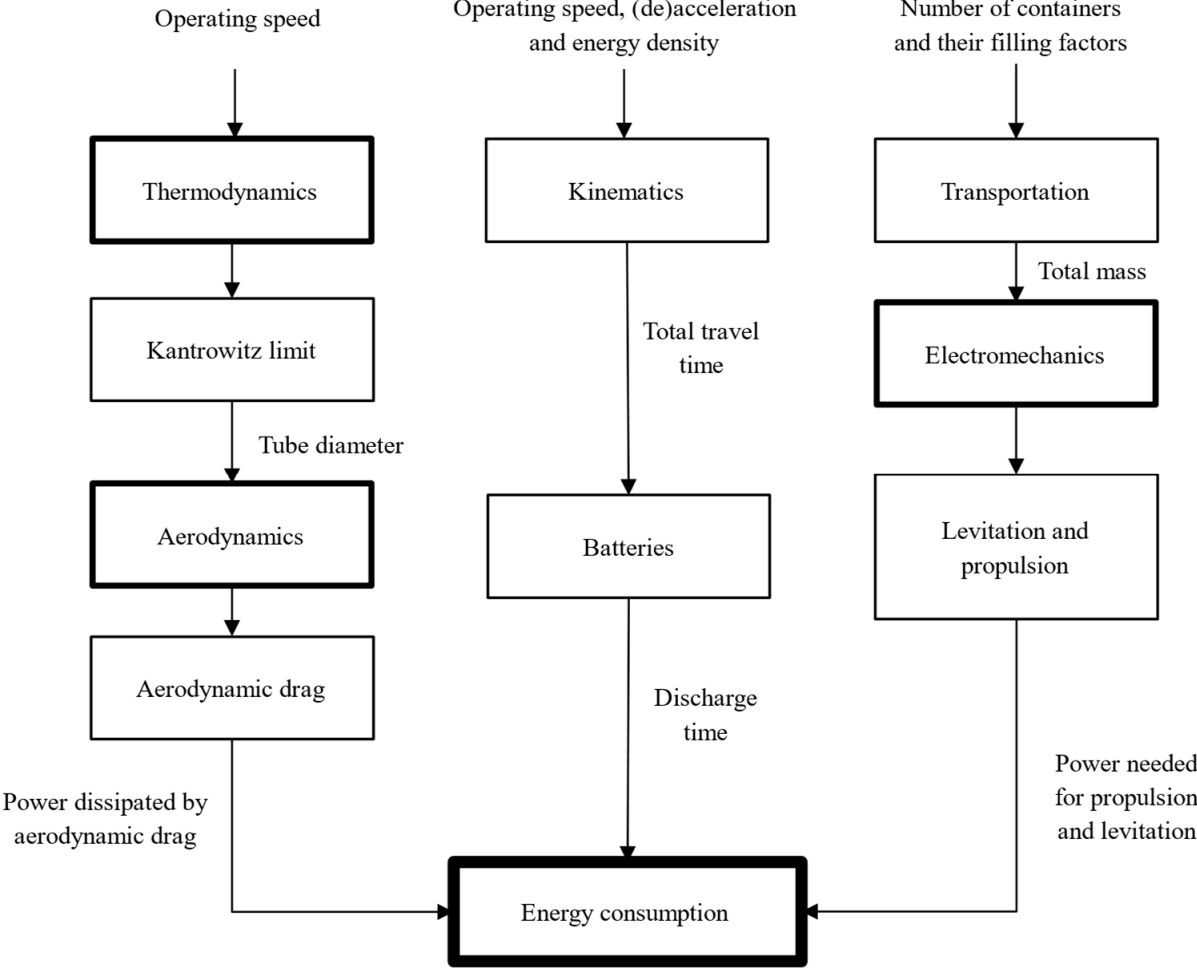

**Figure 1.** Flow diagram of the calculation process (algorithm). Source: Own elaboration.

In this way, it is ensured that the behavior laws of the vehicle inside the tube are fulfilled under all the requirements and considering all the starting hypotheses, with which the physical phenomenon is completely characterized. Once the problem has been formulated, and the behavior equations and the input data are introduced into the software, the software finds the solution to the system of equations. Finally, the design parameters, such as the transported mass, the operating speed, and the length of the tube, are varied according to the simulation procedures. The results (the definitive results of the system of equations) are obtained in relation to the energy consumption of the transport operation.

### 2.3. List of Abbreviations

The list that contains the abbreviations used in the rest of the article can be consulted in Table A1 (Appendix A).

### 2.4. System Definition

The system is defined in this subsection, starting with the system drawings and following with the equations that are to be inserted in the equation blocks shown in Figure 1 (Section 2.2) These blocks are to be presented in alphabetical order.

#### 2.4.1. System Drawings

The next drawings show the system overall. The first two drawings focus on the vehicle and shows the outside and the inside of the capules/pods that compose it, while the third drawing illustrates the electrical model for the considered EDS. Note that Hypotheses (6)–(8), which have to do with the real system geometry or its virtual model, can be visually checked in the drawings:

#### 2.4.2. Aerodynamics

The high-speed transport system runs inside a tube, and this is like a vehicle that runs inside a tunnel, whose drag coefficient increases as a result of the tunnel effect. According to [16], the relation between the drag coefficient inside and outside is expressed as follows (Equation (1)). To calculate the coefficient of drag inside, the same reference includes this formula (Equation (2)). Also, according to this reference, the outside drag coefficient is related to the moment section of the boundary layer (Equation (3)):

$$T_f = \frac{C_{D_t}}{C_{D_{ext}}} \tag{1}$$

$$C_{D_t} = \left( C_{D_{ext}} + \beta \left( \frac{\Delta_1}{A_f} \right)^2 \right) \left( \frac{1 - \frac{c_i}{v}}{1 - \beta \left( \frac{\Delta_1}{A_f} \right)} \right)^2 \tag{2}$$

$$C_{D_{ext}} = 2 \frac{\Delta_2}{A_f} \tag{3}$$

A relationship exists between the boundary layer momentum section and the boundary layer displacement section. To find this relationship, it must be taken into account that the boundary layer will be laminar, as can be verified by calculating both the local and the global Reynolds number, with some data extracted from [24] (first model):

$$Re_{Dc} = \frac{\rho_t v D_c}{\mu_t} \tag{4}$$

$$Re_{Lc} = \frac{\rho_t v L_c}{\mu_t} \tag{5}$$

$$Re_{D_c} = \frac{1.18 \times 10^{-3} \times \frac{1220}{3.60} \times 1.34}{1.80 \times 10^{-5}} = 29,769.51$$

$$Re_{L_c} = \frac{1.18 \times 10^{-3} \times \frac{1220}{3.60} \times 25}{1.80 \times 10^{-5}} = 555,401.23$$

where $1.80 \times 10^{-5}$ Pa·s is the dynamic viscosity for dry air at 20 °C and 100 Pa (the variation of viscosity with pressure is neglectable for such a low pressure) [18]. In addition, 25 m is approximately the length of the capsule, which can be gathered from [24]. The passenger capsule levitates on 28 air bearings, 14 on each side and 1.5 m long each (21 m in total, to which other parts such as the nose and nozzle are added). With respect to $1.18 \times 10^{-3}$ kg/m³, this is the air density and comes from the ideal gas equation.

It can be noted that the local Reynolds is small and not significant, whilst the global can be proper to a laminar boundary layer, since the transition from laminar to turbulent occurs somewhere between $5 \times 10^5$ and $1 \times 10^6$ for a flat plate. Assuming that it is always laminar for the high-speed transportation system, von Karman results can be used to relate the momentum thickness to the displacement thickness through the layer thickness. The process is shown below, after collecting the proper information from [18]: Equations (6) and (7). The function $u(y'')$ could be linear, parabolic, polynomial, etc. As a first approximation, the speed profile is assumed to be linear (Equation (8)). After integrating, the following is obtained (Equations (9) and (10)):

$$\delta^* = \int_0^\delta \left(1 - \frac{u(y'')}{U}\right) dy'' \tag{6}$$

$$\theta = \int_0^\delta \frac{u(y'')}{U} \left(1 - \frac{u(y'')}{U}\right) dy'' \tag{7}$$

$$u(y'') = \frac{U}{\delta} y'' \tag{8}$$

$$\delta^\star = \frac{\delta}{2} \tag{9}$$

$$\theta = \frac{\delta}{6} \tag{10}$$

For the completion of this equation block, the Equations (11)–(16), which come from reference [24], are necessary as well:

$$A_f = A_c \tag{11}$$

$$\beta = \frac{A_c}{A_t} \tag{12}$$

$$\Delta_1 = \frac{\pi}{4} \left(D_{desp}^2 - D_c^2\right) \tag{13}$$

$$\Delta_2 = \frac{\pi}{4} \left(D_{movto}^2 - D_c^2\right) \tag{14}$$

$$D_{movto} = D_c + 2\theta \tag{15}$$

$$D_{desp} = D_c + 2\delta^* \tag{16}$$

### 2.4.3. Electromechanics

For the study of the EDS, the works consulted are [19–21]. The EDS used for this high-speed transportation system is very similar to that used for other magnetic levitation (maglev) vehicles, although in those maglev vehicles wheels are needed at low speeds because there is not enough induction magnetic field to levitate. The traditional EDS can be modeled as a LIM (linear induction motor) for levitation and as an LSM (linear synchronous motor) for propulsion. In order to eliminate the need for wheels, the LIM is replaced by an LSM when applying EDS to the high-speed transportation system, where the rotor will be mounted on the pod (short rotor) and the stator on the tube [21]. From this work, these expressions are taken: Equations (17) and (18). Furthermore, reference [20] contains explanations and formulae for levitation and the drag force generated by the EDS operation. These formulae can be found below, although expressed a little differently (Equations (20)–(22)). Lastly, the next equations from reference [19] have been used in the analysis (Equations (23)–(25)), where the number three indicates the number of phases of the motor:

$$\eta_{EDS} = \frac{F_x v}{F_x v + 3I_1^2 R} \tag{17}$$

$$cos\varphi = \frac{F_x v + 3I_1^2 R}{3V_1 I_1} \tag{18}$$

$$sin\varphi = \frac{X_1 I_1^2 + E_1 I_1 sin\gamma_o}{V_1 I_1} \tag{19}$$

$$F_z = m_{tot} g \tag{20}$$

$$F_{D_{EDS}} = C_{D_{EDS}} F_z \tag{21}$$

$$P_{D_{EDS}} = F_{D_{EDS}} v \tag{22}$$

$$P = 3V_1 I_1 cos\varphi \tag{23}$$

$$Q = 3V_1 I_1 sin\varphi \tag{24}$$

$$3E_1 I_1 cos\gamma_0 = F_x v \tag{25}$$

The electrical model for the considered EDS is shown in [19]. This model is based on the LSM, which can be seen as a rotary synchronous motor rolled out flat. Subsequently, a resistance and a reactance are used at the stator (on the left). At the model air gap, electric power is equated to mechanical power. On the right, a damper and a spring are joined to represent mechanical losses. However, for a first parameter estimation, it is preferable to remove the damper and the spring, and to consider that all active power losses occur in the stator resistance (Figure 2c).

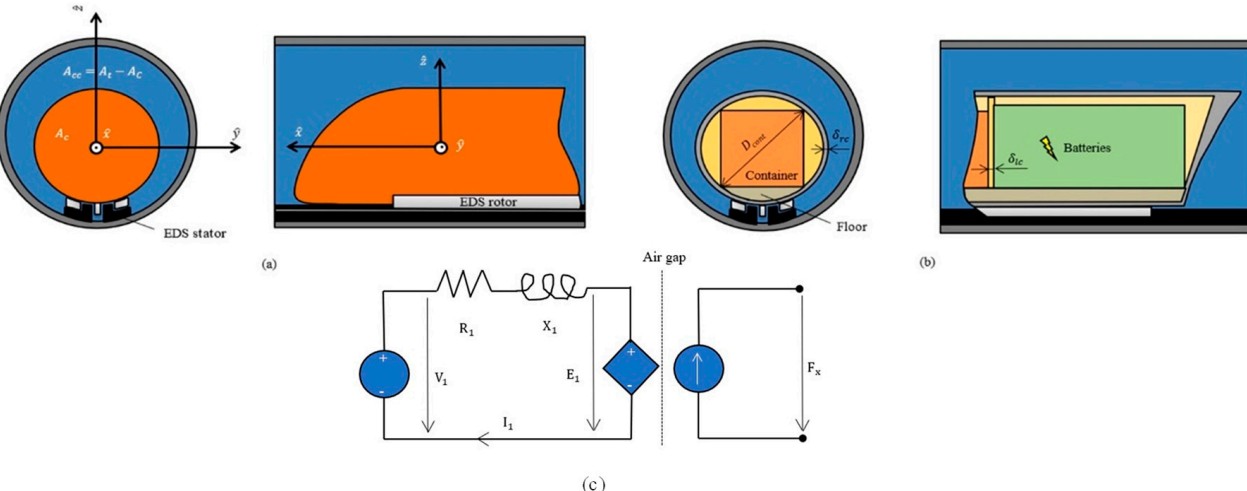

**Figure 2.** (**a**) Cross−sectional drawing of the tube in front of the vehicle and its profile; (**b**) Cross−sectional drawing of the tube and the vehicle, near the rear, and its axial section; (**c**) Electrical model for the considered EDS. Source: Own elaboration.

2.4.4. Thermodynamics

Lastly, to derive the Kantrowitz limit main expression, three basic thermodynamics equations were utilized: Mass flow conservation, Mach number definition, sound speed in an ideal gas, ideal gas law, and isentropic relations for pressure and temperature. The subscript 1 represents the air state or associated variables before the air flows into the annulus and the subscript 2 represents the contrary. In reference [23], most of the aforementioned formulae may be consulted. The main expression to analyze the Kantrowitz limit phenomenon is derived by combining Equations (26)–(32) (subscripts for i = 1 and 2). The complete process can be found in reference [22] and its outcome is Equation (33):

$$\dot{m}_i = \rho_i A_i v_i \tag{26}$$

$$\dot{m}_i = constant \tag{27}$$

$$M_i = \frac{v_i}{a_{s_i}} \tag{28}$$

$$a_{s_i} = \sqrt{\gamma R T_i} \tag{29}$$

$$\rho_i = \frac{p_i}{R T_i} \tag{30}$$

$$\frac{p_{0_t}}{p_i} = \left(1 + \left(\frac{\gamma - 1}{2}\right) M_i\right)^{\frac{\gamma}{\gamma - 1}} \tag{31}$$

$$\frac{T_{0_t}}{T_i} = \left(1 + \left(\frac{\gamma - 1}{2}\right) M_i\right)^{\frac{\gamma}{\gamma - 1}} \tag{32}$$

$$\dot{m}_{cc_{máx}} = A_{cc} \frac{p_{0_t}}{\sqrt{T_{0_t}}} \sqrt{\frac{\gamma}{R}} \left(1 + \left(\frac{\gamma - 1}{2}\right)\right)^{-\left(\frac{\gamma + 1}{2(\gamma - 1)}\right)} \tag{33}$$

Note: $\rho_1 = \rho_t$, $A_1 = A_t$, $A_2 = A_{cc}$, $v_1 = v$. See also Figure 2a,b.

### 2.4.5. Auxiliary Equation Blocks

As shown in Figure 1, the aerodynamics (Equations (1)–(16)), electromechanics (Equations (17)–(25)) and thermodynamics (Equations (26)–(33)) blocks are interrelated through the auxiliary equation blocks. This block comes from reference [24] and comprises the Equations (A1)–(A25). It is presented in Table A2 (Appendix B).

### 2.4.6. Final Equation Block

As shown in Figure 1, the final block of the model is the energy consumption block. This block comes from reference [24] and relies on the results of the rest of the blocks. The final block equations (Equations (34)–(41)) are gathered in Table 1

**Table 1.** Final equation block, coming from reference [24].

| Block | Equation | Left-Side Variable (SI Unit) | Variable Definition | Equation Number |
|---|---|---|---|---|
| Energy consumption | $E_{ac} = \left(\frac{m_{tot} a_1 \frac{v}{2} + \overline{P}_{av}}{\eta_{EDS}}\right) t_{ac}$ | $E_{ac}$ (J) | Energy consumed during acceleration | (34) |
| | $E_{gen} = -\eta_{EDS}\left(m_{tot} a_2 \frac{v}{2} - \overline{P}_{av}\right) t_{dec}$ | $E_{gen}$ (J) | Energy generated during deceleration | (35) |
| | $\overline{P}_{av} = \frac{P_D}{4} + \frac{P_{D_{EDS}}}{3}$ | $\overline{P}_{av}$ (W) | Mean power dissipated by running resistance | (36) |
| | $E_v = \frac{P_{av}}{\eta_{EDS}} t_v$ | $E_v$ (J) | Energy consumed throughout the travel at the speed v | (37) |
| | $E_{bat} = \frac{m_{Li} + e_{bat} \frac{t_{tot}}{t_{des}}}{\eta_{bat}}$ | $E_{bat}$ (J) | Energy consumed by the batteries | (38) |
| | $E_t' = \frac{E_{ac} + E_{gen} + E_v + E_{bat}}{L_t}$ | $E_t'$ (J·m$^{-1}$) | Total energy consumed per unit length | (39) |
| | $e_t' = \frac{E_t'}{m_{carga} \sum_{i=1}^{i=n_{cont}} f_i}$ | $e_t'$ (J·m$^{-1}$·kg$^{-1}$) | Total energy per unit length and payload mass | (40) |
| | $I_e = \frac{E_{ac} + E_{gen} + E_v + E_{bat}}{m_{carga} \sum_{i=1}^{i=n_{cont}} f_i}$ | $I_e$ (J·kg$^{-1}$) | Energy consumption per payload mass (energy index) | (41) |

### 2.5. Software Choice

Once all the equations have been obtained, it is necessary to process them in an equation solver program. Due to the large number of equations and relations that had to be implemented, only software capable of processing the entire volume of data in an agile way has been considered. After considering several options (Mathematica, Matlab

and Engineering Equation Solver), Engineering Equation Solver [25] has been chosen, as it is used in other models that involve thermodynamical equations [26,27]. This program takes the equation blocks along with their inputs and obtains their outputs by means of iterations. These results are obtained after an undetermined number of iterations, depending on adjustable stop criteria such as the relative residuals, which can be as low as $10^{-10}$, or the limit of iterations. The specific version with which the results were obtained is Engineering Equation Solver Professional V9.457-3D (EES). The chosen program, besides solving equations, can create parametric tables and graphs derived from those equations.

*2.6. Simulation Procedures*

The objective is to analyze the capacity of this transport system and compare different alternatives based on their efficiency. However, there is a lot of input data to enter before getting the results, that is, the final values of all the output variables involved.

First, input data are chosen. They may come from different sources: references, calculations, and optimizations with the aid of EES tables and graphs in most cases. Then, they are entered in the program.

Once those data have been selected and entered, the number of containers, speed and tube length can be chosen. The choice of these essential factors that are based on auxiliary factors is what this work focuses on, because they lead to the results. All these results will be obtained for a single vehicle using a single tube, which will be optimized. This vehicle enters the tube, travels through it, and leaves it at the exact instant that a new vehicle begins its journey.

Starting with the number of containers, the most interesting plot to choose is the $I_E - I_C^{-1}$ plot (several curves, one for each number). When selecting it, two factors are key:

1. $I_E$ or, in other words, specific energy consumption to payload, must be the lowest possible.
2. $I_C$ or cargo throughput per unit time must be the highest possible. However, its inverse is used on the plot so that optimal points will fall around the lower-left corner. Seen from another perspective, it can be stated that it is important to minimize the time required to send the payload.

In order to obtain one curve instead of one point with coordinates $(I_C^{-1}, I_E)$ for every number of containers, these two basic variables could be altered:

- Speed, which is a relevant factor, as both $I_E$ and $I_C$ strongly depend on it, so a range of speed values is included as input to make the plot. Were the range not included, then the outcome would be one point with coordinates $(I_C^{-1}, I_E)$ for every number of containers. The range for a high-speed transportation system without a compressor is 700–1000 km/h, as it will be demonstrated later.
- Tube length. As defined in the beginning, it can take one of three discrete values: 500, 750 or 1000 km. $I_E$ and $I_C$ also depend on this to a great extent.

Speed is chosen because the $I_E - I_C^{-1}$ curves as a function of speed will be helpful when selecting it afterwards. Choosing the tube length would not have been useful later because the consumption per unit length would not have been represented.

This leads to the choice of speed. $I_E - I_C^{-1}$ curves are used for this, but Kantrowitz limit results are crucial inasmuch as aerodynamics play a huge role. The speed chosen must comply with the following requirements: working conditions under the Kantrowitz limit while keeping the lowest possible $D_t$, low $I_E$ and high $I_C$ (or low $I_C^{-1}$, its counterpart). Plus, it should leave state-of-the-art maglev speeds behind by a sufficient margin.

The most suitable graph for presenting Kantrowitz limit results is the $D_t - v$ curve. In this way, the speed selected will be the one that optimizes $I_E$, $I_c$ and $D_t$.

After this, the tube length is selected out of the three figures available. This time, $I_E$ is no longer useful on its own. This is because $I_E$ is energy divided by mass, being $E_v$ the factor escalating linearly with $L_t$ Equation (37); and hence through $t_v$ according to Equations (A9)$-$(A13). Were $I_E$ utilized, then 500 km would be optimal for minimizing

both $I_E$ and $I_c^{-1}$, but energy per unit distance would not even have been considered. Energy per unit distance is relevant because it contributes to determine the energy intensity of the operation. With that being said, the unknown $e'_t$ is chosen instead of $I_E$, resulting in $e'_t - I_c^{-1}$ curves. $e'_t$ may be seen as the combination of $I_E$ and $E'_t$ and the optimal length will be the one that minimizes both of them, this being interpreted as pursuing both efficacy and efficiency (that is, effectiveness).

Finally, the optimal values for the number of containers, speed and length are introduced. Once the program has compiled everything, the window with the final values will appear on the screen, arranged in alphabetical order.

### 2.7. Input Data

Firstly, 20′ aluminum Dry Van containers have the following characteristics: 6.058 m ($\cong 20'$) for length ($L_{cont}$), 2.438 m for width, 2.591 m for height, 2180 kg for tare ($m_{tare}$), 28,300 kg for maximum load ($m_{carga}$).

According to the width and height of the container, the parameter $D_{cont}$ is 3.558 m, using Pythagoras' theorem.

After setting the dimensions of the specified container, the rest of the input variables are given values:

1.  $a_1 = a_2 = 14.72$ m/s² (1.5 g). This is because cargo withstands higher accelerations than passengers, as there are not any discomfort issues.
2.  $c_i$ and $g$ are constants and the former is null (there is not any wind flowing inside the tube).
3.  $e_{bat}$, $R$, $\gamma$ and $\eta_{bat}$ were extracted from various references.
4.  The rest were extracted from a reference in which they were optimized.

Table 2 collects all of the input values and provides their references (refs.) except when non−applicable (N/A):

**Table 2.** Input variables with their respective units, their values, and their associated references to their right.

| Variable | Value | Refs. | Variable | Value | Refs. |
|---|---|---|---|---|---|
| $a_1$ (m/s²) | 14.72 | [N/A] | $m_{tara}$ (kg) | 2180 | [24] |
| $a_2$ (m/s²) | 14.72 | [N/A] | $p_t$ (Pa) | 250 | [24] |
| $C_{DEDS}$ (ɸ) | $3 \times 10^{-3}$ | [24] | $R$ (J/(kg·K)) | 287 | [28] |
| $C_{Dext}$ (ɸ) | 0.60 | [24] | $R_1$ (Ω) | 8 | [24] |
| $c_i$ (m/s) | 0 (const.) | [N/A] | $T_t$ (°C) | 20 | [24] |
| $D_{cont}$ (m) | 3.558 | [24] | $\gamma$ (ɸ) | 1.40 | [28] |
| $e_{bat}$ (Wh/kg) | 225 | [29] | $\gamma_o$ (°) | 15 | [24] |
| $g$ (m/s²) | 9.81 (const.) | [N/A] | $\delta_{lc}$ (m) | 0.04 | [24] |
| $L_{cont}$ (m) | 6.058 | [24] | $\delta_{rc}$ (m) | 0.05 | [24] |
| $m_{carga}$ (kg) | 28,300 | [24] | $\eta_{bat}$ (p.u.) | 0.90 | [29] |
| $m_{EB}$ (kg) | * | [24] | $\eta_{EDS}$ (p.u.) | 0.73 | [24] |
| $m'_{EDS}$ (kg/m) | 32 | [24] | $\tau$ (%) | 30 | [24] |
| $m'_{est}$ (kg/m) | 500 | [24] | $\varphi$ (°) | 30 | [24] |
| $m_{Li^+}$ (kg) | * | [24] | | | |

For *: $m_{Li^+} = 350$ kg for $n_{cont} = 1$ and 50 kg is added per each additional container. $m_{EB} = 750$ kg for $n_{cont} = 1$ and 250 kg is added per each additional container, and 350 and 750 kg have been used to start the series.

## 3. Results

### 3.1. $I_E - I_C^{-1}$ Curves

The Figure 3 has been created from the data contained in Table A3 (Appendix C):

In conclusion for Figure 3, when increasing $n_{cont}$ there is an improvement in both $I_E$ and $I_C$, which is clearly smaller after every increment.

When adding one container for the first time, payload (associated with capacity) grows by roughly 30 t. This is a 100 % growth, from 30 to 60 t. When adding one container again,

payload grows by roughly 30 t with respect to the initial 60. This is a 50 % increase. The next time there is a 33 % increase (30/90) and, finally, 25% (30/120). This results in a slowing-pace of increase in $I_C$ (the contrary for $I_C^{-1}$).

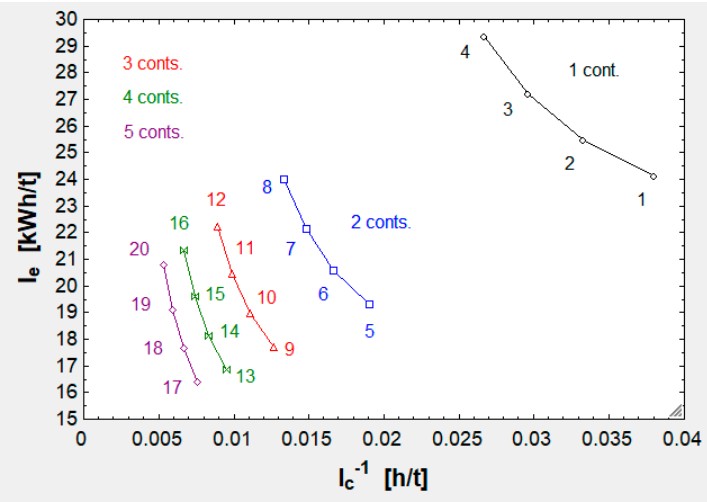

**Figure 3.** $I_E - I_C^{-1}$ curves for $L_t = 750$ km and for 1–5 containers (conts.)

Besides this, the dead weight also grows increment by increment: $m_{Li+}$ and $m_{EB}$ grow as established in Table A3, $m'_{est}$ and $m'_{EDS}$ multiply a longer length ($n_{cont}L_c$ according to Equation (A22)) and $m_{tara}n_{cont}$ according to the same formula. This and the slowing-pace improvement in capacity explains the slowing-pace decrement in $I_E$, which is mainly governed by the ratio $m_{tot} / \left( m_{carga}\sum_{i=1}^{i=n_{cont}} f_i \right)$ (the difference between the numerator and denominator is the deadweight) and by losses independent from $m_{tot}$ (chiefly $P_D t_v$ and $E_{bat}$) divided by payload.

In the end, $n_{cont}$ is set to 5 because the improvement from 5 to 6 will be predictably tinier and over-dimensioning of the system is undesirable.

*3.2. $D_t - v$ Curve*

The Figure 4 has been created from the data contained in Table A4 (Appendix C), focusing on the speed range 500–1000 km/h so as to facilitate the analysis:

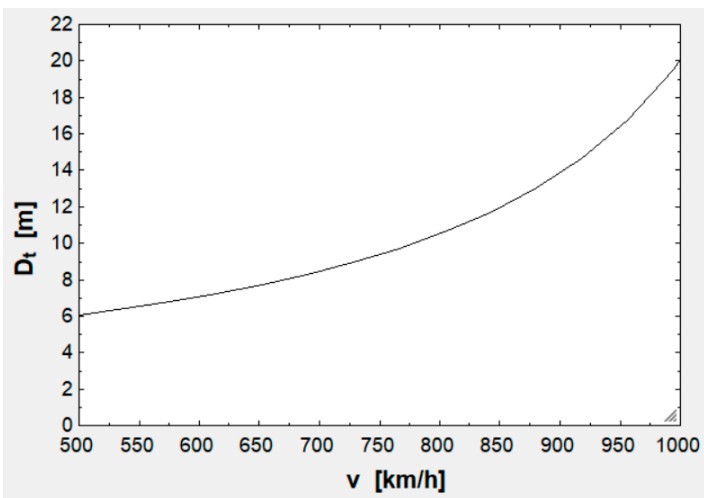

**Figure 4.** $D_t - v$ curve.

Analyzing Figure 4, it can be deduced that the zone of interest goes from 700 to 800 km/h ($D_t$ around 9 m), for the following reasons: 9 m is suitable considering that $D_c$ is 3.658 m, so that blockage will be small (0.16 or 16% at 728 km/h, according to Table A4); speeds below 700 are near state-of-the-art maglev speeds and speeds above 800 yield a $D_t$ rising at a higher rate.

The relevant information provided by Figure 4 concerning v is that the ends of any speed range should be avoided: lower speeds yield a low $I_E$, but low $I_C$ (or high $I_C^{-1}$). By contrast, the highest speeds imply the contrary. This means that the optimal speed will be near the center of the speed interval.

This being said, $v$ is chosen as 750 km/h.

### 3.3. $e_t' - I_C^{-1}$ Curves

The Figure 5 has been created from the data contained in Table A5 (Appendix C):

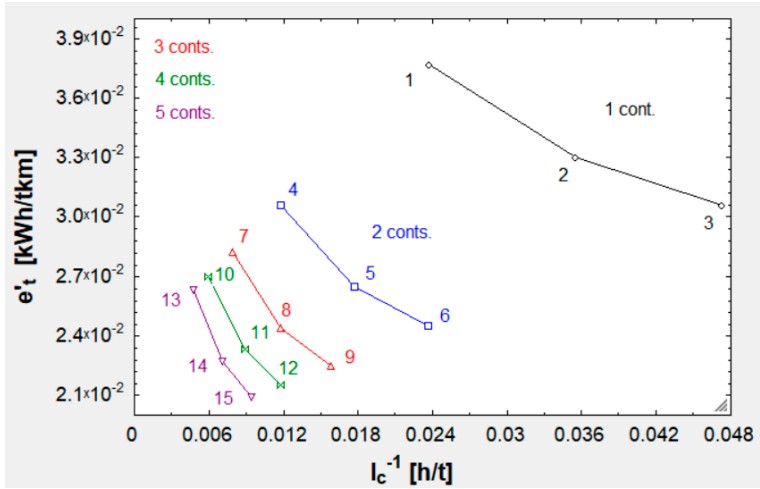

**Figure 5.** $e_t' - I_C^{-1}$ curves at $v = 750$ km/h and for 1–5 conts.

In Figure 5, and in contrast to the $I_E - I_C^{-1}$ curves (Figure 3), now $e_t'$ has replaced $I_E$. It must be noted that $e_t'$ can be calculated as $E_t'$ divided by $m_{tot}$ or $I_E$ divided by $L_t$. This means that all of the tendencies observed before are still valid. Now there are two additional tendencies, explained next.

In the first place, $E_t'$ decays as $L_t$ augments, as Table A5 proves. This is due to the fact that accelerating the vehicle requires the provision of a high amount of kinetic energy and this energy is better used for longer routes.

Secondly, $I_C$ worsens as $L_t$ grows. It is simple to understand this by reviewing Equations (A12) and (A25): as $L_t$ grows, $t_{tot}$ does too and $I_C$ decreases (or $I_C^{-1}$ increases). Shorter routes allow a higher throughput because, for the same period of time, more containers can be dispatched.

After having seen the different trends involved, it can be concluded that the best option is $L_t = 750$ km. 750 km (point/run 14 above in Figure 5 and Table A5) is the only one that optimizes $e_t'$ (associated with both $I_E$ and $E_t'$) and $I_C$. In addition, 500 km (point/run 13) improves $I_C$ and its counterpart but worsens $e_t'$, while 1000 km (point/run 15) has the contrary effect.

### 3.4. Definitive Results

Once the optimized parameters ($n_{cont} = 5$, $v = 750$ km/h and $L_t = 750$ km) have been introduced, EES solves through the whole equation system (the algorithm), providing the definitive results. The operating point of this vehicle is shown below, in Figure 6:

At this operating point, the specific values adopted by all of the variables are collected in Table 3 and are discussed in Section 4:

**Table 3.** Variables with their respective units and their values to their right.

| Variable | Value | Variable | Value |
|---|---|---|---|
| $A_c$ (m$^2$) | 10.51 | $m_{EB}$ (kg) | 1750 |
| $A_{cc}$ (m$^2$) | 58.38 | $m'_{EDS}$ (kg/m) | 32 |
| $A_f$ (m$^2$) | 10.51 | $m'_{est}$ (kg/m) | 500 |
| $A_t$ (m$^2$) | 68.89 | $m_{Li^+}$ (kg) | 550 |
| $a_1$ (m/s$^2$) | 14.72 | $\dot{m}_t$ (kg/s) | 42.64 |
| $a_2$ (m/s$^2$) | 14.72 | $m_{tara}$ (kg) | 2180 |
| $a_s$ (m/s) | 343.20 | $m_{tot}$ (kg) | 171,027 |
| $C_{DEDS}$ ($\phi$) | $3 \times 10^{-3}$ | $n_{cont}$ ($\phi$) | 5 |
| $C_{Dext}$ ($\phi$) | 0.60 | $P_1$ (MW) | 720 |
| $C_{Dt}$ ($\phi$) | 1.07 | $P_{av}$ (MW) | 1.20 |
| $c_I$ (m/s) | 0 | $P_D$ (W) | 150,991 |
| $D_c$ (m) | 3.658 | $P_{D_{EDS}}$ (MW) | 1.05 |
| $D_{carga}$ (m) | 3.558 | $P_x$ (MW) | 526 |
| $D_{cont}$ (m) | 3.558 | $p_t$ (Pa) | 250 |
| $D_{desp}$ (m) | 5.20 | $p_{o_t}$ (Pa) | 320.65 |
| $D_{movto}$ (m) | 4.17 | $R$ (J/kg·K) | 287 |
| $D_t$ (m) | 9.37 | $R_{av}$ (N) | 5758 |
| $E_1$ (V) | 63,736 | $R_1$ ($\Omega$) | 8 |
| $E_{ac}$ (kWh) | 1414.39 | $T_f$ ($\phi$) | 1.78 |
| $E_{bat}$ (kWh) | 105.77 | $T_t$ (°C) | 20 |
| $E_{gen}$ (kWh) | $-751.50$ | $T_{o_t}$ (°C) | 41.60 |
| $E'_t$ (kWh/km) | 3.21 | $t_{ac}$ (s) | 14.15 |
| $E_v$ (kWh) | 1636.83 | $t_{dec}$ (s) | 14.15 |
| $e_{bat}$ (Wh/kg) | 225 | $t_{des}$ (min) | 78.31 |
| $e'_t$ (kWh/tkm) | $2.27 \times 10^{-2}$ | $t_{tot}$ (min) | 60.24 |
| $F_D$ (N) | 724.76 | $t_v$ (min) | 59.76 |
| $F_{D_{EDS}}$ (N) | 5033.33 | $V_1$ (V) | 97,381 |
| $F_x$ (MN) | 2.52 | $v$ (km/h) | 750 |
| $F_z$ (MN) | 1.68 | $X_1$ ($\Omega$) | 11.31 |
| $f_1, \dots, f_5$ ($\phi$) | 1 | $\beta$ (%) | 15.26 |
| $g$ (m/s$^2$) | 9.81 | $\gamma$ ($\phi$) | 1.40 |
| $I_1$ (A) | 2846 | $\gamma_o$ (°) | 15 |
| $I_c$ (t/h) | 140.95 | $\Delta_1$ (m$^2$) | 10.70 |
| $I_e$ (kWh/t) | 17 | $\Delta_2$ (m$^2$) | 3.15 |
| $L_{ac}$ (km) | 1.47 | $\delta^*$ (m) | 0.77 |
| $L_c$ (m) | 6.138 | $\delta_{lc}$ (m) | 0.04 |
| $L_{cont}$ (m) | 6.058 | $\delta_{rc}$ (m) | 0.05 |
| $L_{dec}$ (km) | 1.47 | $\eta_{bat}$ (p.u.) | 0.90 |
| $L_t$ (km) | 750 | $\eta_{EDS}$ (p.u.) | 0.73 |
| $L_v$ (km) | 747.05 | $\theta$ (m) | 0.26 |
| $M$ ($\phi$) | 0.61 | $\rho_t$ (kg/m$^3$) | $2.97 \times 10^{-3}$ |
| $m_{carga}$ (kg) | 28,300 | $\tau$ (%) | 30 |
| $\dot{m}_{cc}$ (kg/s) | 42.64 | $\varphi$ (°) | 30 |

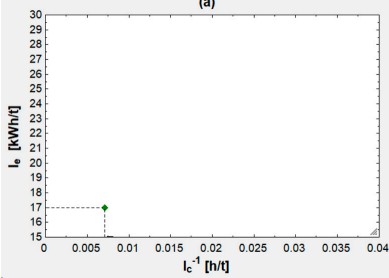 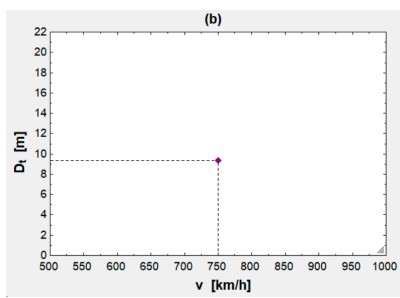 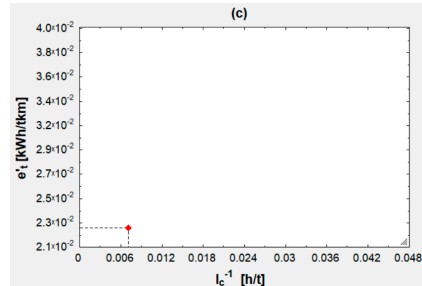

**Figure 6.** Operating point of the predesigned vehicle. (**a**) (18.23 kWh/t; $8.87 \times 10^{-3}$ h/t). (**b**) (9.37 m; 750 km/h). (**c**) ($2.27 \times 10^{-2}$ kWh/tkm; $8.87 \times 10^{-3}$ h/t).

## 4. Discussion

A high-speed transportation system (Hyperloop) for freight transport has been pre-designed in this work. As seen in Figure 6, it has been found that the most effective configuration is a vehicle with 5 containers moving at 750 km/h in a 750-km tube. This vehicle is capable of delivering 141 t/h with a consumption of $2.27 \times 10^{-2}$ kWh/tkm.

However, it is necessary to review certain aspects found when analyzing the predesign (the values from Table 3):

- $D_t$ measures 9.37 m and is too large because this vehicle lacks an instrument (such as a compressor) to bypass the incoming air. This also impacts speed, since a high-speed transportation system equipped with a compressor is able to reach higher speeds. This last case, in which the vehicle levitated on air bearings instead of magnets, was studied in [24] as an alternative and it was found that this type of vehicle cannot transport a huge amount of freight because maximum mass flow through the compressor limits pressure under the air bearings and, therefore, payload. Nonetheless, a high-speed transportation system with a compressor levitating on magnets has not yet been studied and is proposed for further works. Moreover, it should also be studied whether, or not, it is feasible to build a 9-m diameter tube.

- It should be determined if it is feasible to build a 750 km long EDS or if it would be preferrable to build only some sections of it (the vehicle would cruise between those sections, as proposed by [17]).

- At the ends, near the cargo terminals (at the acceleration and deceleration lengths), this EDS system should be able to withstand 720 MW ($P_1$ in Table 3) and evacuate the generated heat (194.40 MW, 27%, because $1 - \eta_{EDS} = 0.27$). State-of-the-art EDS systems are not likely to withstand such an enormous power. In this case, the first approximation done in this work should be discarded and acceleration reduced. If, for instance, 2.10 m/s$^2$ (1/7 of 14.72) were to be used, its power would be 103 MW, a result which nears the result of reference [17]: 50 MW is the traction power for his first model [17] and for his second model it is 87 MW. The mean speed would barely be affected, as it would only be reduced from 747 to 730 km/h. This can be previewed on the next plot (Figure 7), although the optimization of acceleration and EDS power should be deeply studied, so they are two topics proposed for further studies:

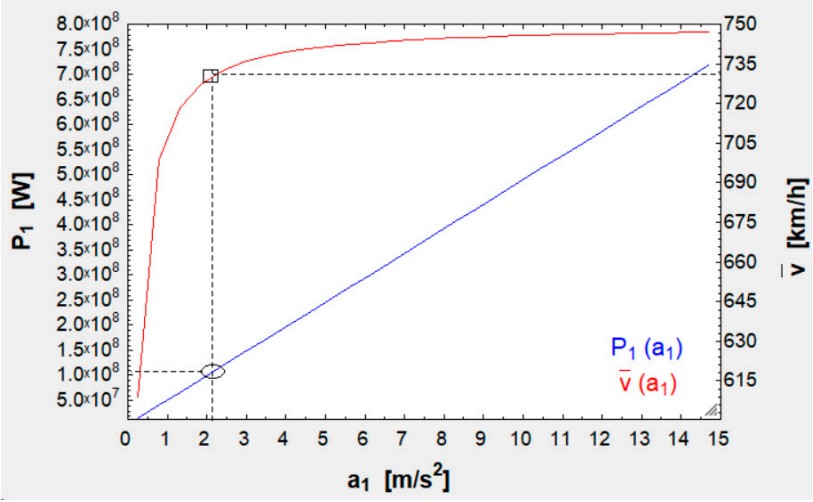

**Figure 7.** Correlations between $a_1$ and $P_1$ and between $a_1$ and $\bar{v}$. At 2.0 m/s$^2$, $P_1 = 103$ MW and $\bar{v} = 730$ km/h.

- In addition to the optimization of EDS power, the pole pitch should be adjusted to have $C_{D_{EDS}} = 3 \times 10^{-3}$ or even lower. Thanks to such a good coefficient $F_{D_{EDS}}$ is only

5 kN (very low in comparison with $F_z$, which is greater than 1.5 MN), and this drag force can further be lowered, as is shown in Figure 8:

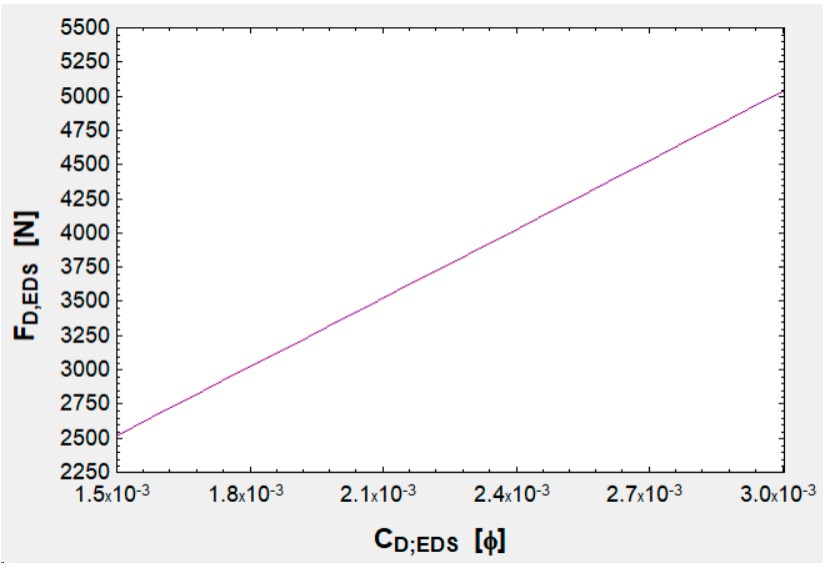

**Figure 8.** Correlation between $C_{D,\ EDS}$ and $F_{D,EDS}$, the latter 5 kN at $C_{D_{EDS}} = 3 \times 10^{-3}$.

- An attempt to lower $m'_{est}$, which is the main dead weight of the vehicle, could also be carried out. Nowadays, there are many composite materials reinforced with advanced fibers (carbon-graphite, aramid, etc.) and alloys (aluminum alloys, for instance) which could lighten the vehicle while resisting the stresses and forces generated. If the structure weight could be lowered by 50%, then the energy index could be cut by 3.88% (Figure 9a), and the energy saving could be further augmented to 4.59% by cutting the rest of dead weight (namely, $m'_{EDS}$ and $m_{EB}$) in half also (Figure 9b):

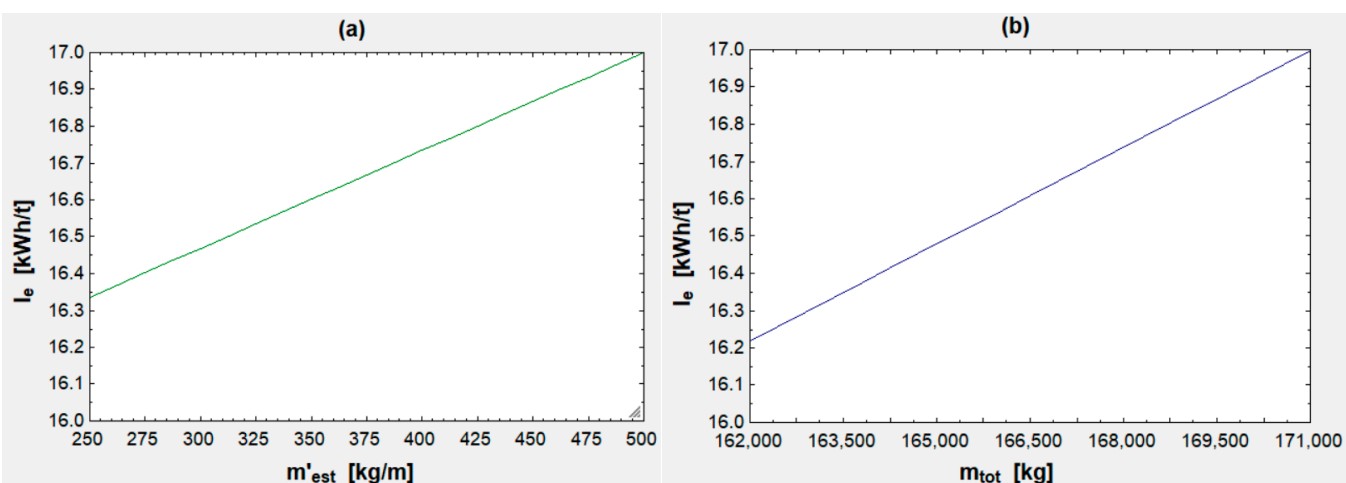

**Figure 9.** (**a**) Correlation between $m'_{est}$ and $I_e$. (**b**) Correlation between $m_{tot}$ and $I_e$. The latter is 17 kWh/t at $m'_{est} = 500$ kg/m and can be reduced by eliminating some dead weight, as indicated in the text.

- With respect to the Kantrowitz limit, the vehicle is meant to run at 750 km/h in a 9.37-m diameter tube and at 20 °C. At 30 °C, speed could be slightly raised and at 10 °C speed should be slightly diminished to avoid the air stacking. These possible slight variations are illustrated in Figure 10:

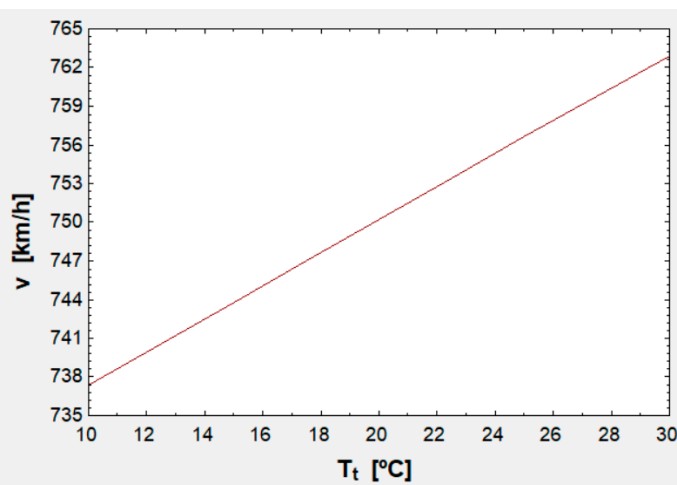

**Figure 10.** Correlation between $T_t$ and $v$. At the operating point, $T_t = 20\,°C$ and $v = 750\,\mathrm{km/h}$ and when $T_t$ varies, the operating speed can be raised or lowered.

- In relation with the previous point, it must be noted that, if this high-speed transportation system were to run in extremely high or low temperature environments, then speed should be diminished or augmented, respectively, although a redesigning process would be more efficient.
- As for the feasibility of keeping airtightness at 250 Pa, it should also be studied, though it is presumably more feasible than 100 Pa, the pressure proposed by [17]. In relation to this, another proposal is the improvement of $C_{D_{ext}}$ through CFD simulation or a wind tunnel.

Finally, it is worth discussing that the current work could be extended to maglev vehicles that run in the open air, this being the main difference in regard to the Hyperloop system. As explained in reference [24], these maglev vehicles rely their levitation and propulsion on either electromechanical suspensions (EMS) or electrodynamic suspensions (EDS) that need wheels for levitation at low speeds (because they are based on a LIM instead of a LSM, as explained in Section 1). In addition, these vehicles incorporate an energy generating unit which transforms some of the input power to the power that feeds the on-board systems, which constitutes an additional resistance [24]. As a result, the algorithm for open-air maglev vehicles would be similar overall but with some differences; for example, the need to calculate thermodynamics and the Kantrowitz limit would be reconsidered, and the rest of the blocks corresponding to aerodynamics, electromechanics, levitation and propulsion could be simplified or expanded according to each case. Likewise, a block on on-board energy generation should be included, interrelating it with that of batteries.

## 5. Conclusions

Through the mathematical modeling of a novel high-speed transport system based on the use of vacuum tubes, the most convenient design has been obtained to allow an effective freight transport operation, which is also efficient in terms of energy. This effective freight transport operation complies with all of the technical requirements and with all the limitations of the physical problem.

The model allows taking into account all the equations involved by the electromechanical, aerodynamic, and thermodynamic laws present in the definition of the problem. By introducing boundary conditions and starting hypotheses, the model allows an analysis of parametric variation to be carried out.

In the case presented, the optimal number of containers that can be transported at high speed with the lowest possible energy consumption can be obtained as a result, in a technically feasible model.

As a continuation of the research work, the next steps to be carried out will consist of the consideration of solving the problem with the restrictions and difficulties that come with using a tube with different curvatures as infrastructure, and with the existence of slopes along the route.

**Author Contributions:** Conceptualization, D.S.P. and E.L.; methodology, D.S.P.; software, D.S.P.; validation, D.S.P. and E.L.; formal analysis, D.S.P.; investigation, D.S.P.; resources, D.S.P. and E.L.; data curation, E.L.; writing—original draft preparation, D.S.P.; writing—review and editing, E.L.; visualization, D.S.P.; supervision, E.L.; project administration, E.L.; funding acquisition, E.L. All authors have read and agreed to the published version of the manuscript.

**Funding:** This research received no external funding.

**Data Availability Statement:** No new data were created or analyzed in this study. Data sharing is not applicable to this article.

**Acknowledgments:** This research work is a summary of the bachelor thesis previously completed by the authors.

**Conflicts of Interest:** The authors declare no conflicts of interest.

## Appendix A

**Table A1.** Abbreviations.

| Abbreviation | Definition | Unit (SI) | Abbreviation | Definition | Unit (SI) |
|---|---|---|---|---|---|
| $A_c$ | Pod cross-sectional area | m$^2$ | $m_{EB}$ | Emergency brakes mass | kg |
| $A_{cc}$ | Annulus area | m$^2$ | $m'_{EDS}$ | EDS magnets mass per unit length | kg·m$^{-1}$ |
| $A_f$ | Frontal area projected on a plane normal to the tube | m$^2$ | $m'_{est}$ | Structural mass per unit length | kg·m$^{-1}$ |
| $A_t$ | Tube cross-sectional area | m$^2$ | $m_{Li^+}$ | Batteries mass | kg |
| $a_1$ | Acceleration | m·s$^{-2}$ | $\dot{m}_t$ | Mass flow through the tube (relative to vehicle) | kg·s$^{-1}$ |
| $a_2$ | Deceleration | m·s$^{-2}$ | $m_{tara}$ | Tare of one container | kg |
| $a_s$ | Sound speed | m·s$^{-1}$ | $m_{tot}$ | Vehicle total mass | kg |
| $C_{DEDS}$ | EDS drag coefficient | ϕ | $n_{cont}$ | Number of containers transported | ϕ |
| $C_{Dext}$ | Drag coefficient outside the tube | ϕ | $P_1$ | Input power to EDS | W |
| $C_{Dt}$ | Drag coefficient inside the tube | ϕ | $P_{av}$ | Power dissipated by running resistance | W |
| $c_i$ | Wind speed induced inside the tube | m·s$^{-1}$ | $P_D$ | Power dissipated by aerodynamic drag | W |
| $D_c$ | Capsule diameter | m | $P_{D_{EDS}}$ | Power dissipated by EDS drag | W |
| $D_{carga}$ | Diameter needed to fit the cargo | m | $P_x$ | Power really used for propulsion | W |
| $D_{cont}$ | Diameter of the circumference surrounding one container | m | $p_t$ | Pressure inside the tube | Pa |
| $D_{desp}$ | Displacement diameter | m | $p_{o_t}$ | Total pressure inside the tube | Pa |
| $D_{movto}$ | Momentum diameter | m | $R$ | Constant for a certain ideal gas | J·kg$^{-1}$·K$^{-1}$ |
| $D_t$ | Tube diameter | m | $R_{av}$ | Vehicle running resistance | N |

**Table A1.** *Cont.*

| Abbreviation | Definition | Unit (SI) | Abbreviation | Definition | Unit (SI) |
|---|---|---|---|---|---|
| $E_1$ | Phase voltage at the stator after losses | V | $R_1$ | Stator resistance | $\Omega$ |
| $E_{ac}$ | Energy consumed during acceleration | J | $T_f$ | Tunnel factor | $\phi$ |
| $E_{bat}$ | Energy consumed by the batteries | J | $T_t$ | Temperature inside the tube | K |
| $E_{gen}$ | Energy generated during deceleration | J | $T_{o_t}$ | Total temperature inside the tube | K |
| $E'_t$ | Total energy consumed per unit length | $J \cdot m^{-1}$ | $t_{ac}$ | Acceleration time | s |
| $E_v$ | Energy consumed throughout the travel at the speed v | J | $t_{dec}$ | Deceleration time | s |
| $e_{bat}$ | Battery stored energy per unit mass | $J \cdot kg^{-1}$ | $t_{des}$ | Batteries discharge time | s |
| $e'_t$ | Total energy per unit length and payload mass | $J \cdot m^{-1} \cdot kg^{-1}$ | $t_{tot}$ | Total route time | s |
| $F_D$ | Drag force | N | $t_v$ | Travel time at the speed v | s |
| $F_{D_{EDS}}$ | EDS drag force | N | $V_1$ | Phase input voltage to the stator | V |
| $F_x$ | Propulsion force (along x axis) | N | $v$ | Vehicle operating speed | $m \cdot s^{-1}$ |
| $F_z$ | Levitation force (along z axis) | N | $X_1$ | Stator reactance | $\Omega$ |
| $f_i$ | Filling factor of each container (for $i = 1, 2, \ldots, n_{cont}$) | $\phi$ | $\beta$ | Blockage ratio | $\phi$ |
| $g$ | Gravity acceleration | $m \cdot s^{-2}$ | $\gamma$ | Adiabatic index | $\phi$ |
| $I_1$ | Stator line current | A | $\gamma_o$ | Angle between $E_1$ e $I_1$ | rad |
| $I_c$ | Transport capacity per unit time (capacity index) | $kg \cdot s^{-1}$ | $\Delta_1$ | Displacement section | $m^2$ |
| $I_e$ | Energy consumption per payload mass (energy index) | $J \cdot kg^{-1}$ | $\Delta_2$ | Momentum section | $m^2$ |
| $L_{ac}$ | Acceleration length | m | $\delta^*$ | Boundary layer displacement thickness | m |
| $L_c$ | Length of one capsule | m | $\delta_{lc}$ | Pod longitudinal thickness | m |
| $L_{cont}$ | Length of one container | m | $\delta_{rc}$ | Pod radial thickness | m |
| $L_{dec}$ | Deceleration length | m | $\eta_{bat}$ | Battery charging efficiency | $\phi$(p.u.) |
| $L_t$ | Tube length (same as the route one) | m | $\eta_{EDS}$ | EDS efficiency | $\phi$(p.u.) |
| $L_v$ | Travel length at the speed v | m | $\theta$ | Boundary layer momentum thickness | m |
| $M$ | Mach number | $\phi$ | $\rho_t$ | Density inside the tube | $kg \cdot m^{-3}$ |
| $m_{carga}$ | Maximum cargo of one container | kg | $\tau$ | Percentage of battery duration over travel time | $\phi$(%) |
| $\dot{m}_{cc}$ | Mass flow through the annulus | $kg \cdot s^{-1}$ | $\varphi$ | EDS power angle | rad |

## Appendix B

**Table A2.** Auxiliary equation blocks, all of which come from reference [24].

| Block | Equation | Left–Side Variable [SI Unit] | Variable Definition | Equation Number |
|---|---|---|---|---|
| Kantrowitz limit | $A_{cc} = A_t - A_c$ | $A_{cc}$ (m²) | Annulus area | (A1) |
| | $A_t = \frac{\pi}{4} D_t^2$ | $A_t$ (m²) | Tube cross-sectional area | (A2) |
| | $A_c = \frac{\pi}{4} D_c^2$ | $A_c$ (m²) | Pod cross-sectional area | (A3) |
| | $D_c = D_{carga} + 2\delta_{rc}$ | $D_c$ (m²) | Capsule diameter | (A4) |
| | $\dot{m}_t = \dot{m}_{cc\,máx}$ | $\dot{m}_t$ (kg·s⁻¹) | Mass flow through the tube (relative to vehicle) | (A5) |
| Aerodynamic drag | $F_D = \frac{1}{2}\rho_t v^2 A_f T_f C_{D_{ext}}$ | $F_D$ (N) | Drag force | (A6) |
| | $P_D = F_D v$ | $P_D$ (W) | Power dissipated by aerodynamic drag | (A7) |
| Batteries | $t_{des} = \left(1 + \frac{\tau}{100}\right) t_{tot}$ | $t_{des}$ (s) | Batteries discharge time | (A8) |
| Kinematics | $t_{ac} = \frac{v}{a_1}$ | $t_{ac}$ (s) | Acceleration time | (A9) |
| | $t_{dec} = \frac{v}{a_2}$ | $t_{dec}$ (s) | Deceleration time | (A10) |
| | $\bar{v} = \frac{\frac{v}{2}(t_{ac}+t_{dec})+vt_v}{t_{ac}+t_{dec}+t_v}$ | $\bar{v}$ (m·s⁻¹) | Mean speed of the vehicle | (A11) |
| | $t_{tot} = \frac{L_t}{\bar{v}}$ | $t_{tot}$ (s) | Total route time | (A12) |
| | $t_v = t_{tot} - t_{ac} - t_{dec}$ | $t_v$ (s) | Travel time at the speed v | (A13) |
| | $L_{ac} = \frac{v^2}{2a_1}$ | $L_{ac}$ (m) | Acceleration length | (A14) |
| | $L_{dec} = \frac{v^2}{2a_2}$ | $L_{dec}$ (m) | Deceleration length | (A15) |
| | $L_v = L_t - L_{ac} - L_{dec}$ | $L_v$ (m) | Travel length at the speed v | (A16) |
| Levitation and propulsion | $F_x = m_{tot}a_1 + R_{av}$ | $F_x$ (N) | Propulsion force (along x axis) | (A17) |
| | $P_x = F_x v$ | $P_x$ (W) | Power really used for propulsion | (A18) |
| | $R_{av} = F_D + F_{D_{EDS}}$ | $R_{av}$ (N) | Vehicle running resistance | (A19) |
| | $P_{av} = R_{av} v$ | $P_{av}$ (W) | Power dissipated by running resistance | (A20) |
| | $P_1 = \frac{F_x v}{\eta_{EDS}}$ | $P_1$ (W) | Input power to EDS | (A21) |
| Transportation | $m_{tot} = n_{cont} L_c \left(m'_{est} + m'_{EDS}\right) +$ $+ m_{Li^+} + m_{EB} +$ $+ m_{carga}\sum_{i=1}^{i=n_{cont}} f_i + n_{cont} m_{tara}$ | $m_{tot}$ (kg) | Vehicle total mass | (A22) |
| | $D_{carga} = D_{cont}$ | $D_{carga}$ (m) | Diameter needed to fit the cargo | (A23) |
| | $L_c = L_{cont} + 2\delta_{lc}$ | $L_c$ (m) | Length of one capsule | (A24) |
| | $I_c = \frac{m_{carga}\sum_{i=1}^{i=n_{cont}} f_i}{t_{tot}}$ (Note that Equation (A25) is not the traditional capacity equation. This Equation has been specifically engineered for this problem, assuming that only one vehicle is using the tube at a time, i.e., the one which is to be optimized.) | $I_c$ (kg·s⁻¹) | Transport capacity per unit time (capacity index) | (A25) |

## Appendix C

The following table contains the parameters that were given values to obtain the graphs. $v$ is given four values: 700, 800, 900 and 1000 km/h. The rest of the input data was also compiled by the program. It should be noted that all of the power systems are predesigned for the maximum possible payload ($f_i = 1$ for i = 1, 2, ..., 5) because it is the worst-case scenario for the EDS and the power system:

**Table A3.** Parameters given values to obtain the output variables and the plot. The output variables are shown in italics: $m_{tot}$ is for consultation and $I_C$, $I_E$ for the curves.

| Run | $n_{cont}$ ($\phi$) | $f_1$ ($\phi$) | $f_2$ ($\phi$) | $f_3$ ($\phi$) | $f_4$ ($\phi$) | $f_5$ ($\phi$) | $v$ (km/h) | $m_{Li^+}$ (kg) | $m_{EB}$ (kg) | $m_{tot}$ (kg) | $I_C$ (t/h) | $I_E$ (kWh/t) | $I_C^{-1}$ (h/t) |
|---|---|---|---|---|---|---|---|---|---|---|---|---|---|
| 1 | 1 | 1 | 0 | 0 | 0 | 0 | 700 | 350 | 750 | 34,845 | 26.32 | 24.11 | $3.80 \times 10^{-2}$ |
| 2 | 1 | 1 | 0 | 0 | 0 | 0 | 800 | 350 | 750 | 34,845 | 30.05 | 25.46 | $3.33 \times 10^{-2}$ |
| 3 | 1 | 1 | 0 | 0 | 0 | 0 | 900 | 350 | 750 | 34,845 | 33.77 | 27.19 | $2.96 \times 10^{-2}$ |
| 4 | 1 | 1 | 0 | 0 | 0 | 0 | 1000 | 350 | 750 | 34,845 | 37.47 | 29.36 | $2.67 \times 10^{-2}$ |
| 5 | 2 | 1 | 1 | 0 | 0 | 0 | 700 | 400 | 1000 | 68,891 | 52.65 | 19.29 | $1.90 \times 10^{-2}$ |
| 6 | 2 | 1 | 1 | 0 | 0 | 0 | 800 | 400 | 1000 | 68,891 | 60.10 | 20.58 | $1.66 \times 10^{-2}$ |
| 7 | 2 | 1 | 1 | 0 | 0 | 0 | 900 | 400 | 1000 | 68,891 | 67.54 | 22.15 | $1.48 \times 10^{-2}$ |
| 8 | 2 | 1 | 1 | 0 | 0 | 0 | 1000 | 400 | 1000 | 68,891 | 74.94 | 24.01 | $1.33 \times 10^{-2}$ |
| 9 | 3 | 1 | 1 | 1 | 0 | 0 | 700 | 450 | 1250 | 102,936 | 78.97 | 17.68 | $1.27 \times 10^{-2}$ |
| 10 | 3 | 1 | 1 | 1 | 0 | 0 | 800 | 450 | 1250 | 102,936 | 90.16 | 18.95 | $1.11 \times 10^{-2}$ |
| 11 | 3 | 1 | 1 | 1 | 0 | 0 | 900 | 450 | 1250 | 102,936 | 101.31 | 20.46 | $9.87 \times 10^{-3}$ |
| 12 | 3 | 1 | 1 | 1 | 0 | 0 | 1000 | 450 | 1250 | 102,936 | 112.41 | 22.23 | $8.90 \times 10^{-3}$ |
| 13 | 4 | 1 | 1 | 1 | 1 | 0 | 700 | 500 | 1500 | 136,986 | 105.29 | 16.88 | $9.50 \times 10^{-3}$ |
| 14 | 4 | 1 | 1 | 1 | 1 | 0 | 800 | 500 | 1500 | 136,986 | 120.21 | 18.14 | $8.32 \times 10^{-3}$ |
| 15 | 4 | 1 | 1 | 1 | 1 | 0 | 900 | 500 | 1500 | 136,986 | 135.08 | 19.62 | $7.40 \times 10^{-3}$ |
| 16 | 4 | 1 | 1 | 1 | 1 | 0 | 1000 | 500 | 1500 | 136,986 | 149.89 | 21.33 | $6.67 \times 10^{-3}$ |
| 17 | 5 | 1 | 1 | 1 | 1 | 1 | 700 | 550 | 1750 | 171,027 | 131.62 | 16.40 | $7.60 \times 10^{-3}$ |
| 18 | 5 | 1 | 1 | 1 | 1 | 1 | 800 | 550 | 1750 | 171,027 | 150.26 | 17.65 | $6.66 \times 10^{-3}$ |
| 19 | 5 | 1 | 1 | 1 | 1 | 1 | 900 | 550 | 1750 | 171,027 | 168.84 | 19.12 | $5.92 \times 10^{-3}$ |
| 20 | 5 | 1 | 1 | 1 | 1 | 1 | 1000 | 550 | 1750 | 171,027 | 187.36 | 20.80 | $5.34 \times 10^{-3}$ |

In the table below, only the parameter v is given values. This is because the rest of the values are either constants or optimized ones. The lower limit is 500 km/h, a speed reachable by state-of-the-art maglev or even high-speed vehicles. The upper one is 1222 km/h, around the 1220 km/h proposed by [24]. At 20 °C and with $\gamma = 1.40$ and $R = 287$ J/(kg·K), $a_s$ equals 1235.53 km/h (by means of Equation (29)), which is slightly superior to 1222 km/h and means that, even if the speed were that, the vehicle would not break the sound barrier and the first hypothesis would still be true:

**Table A4.** First, values given to $v$. Second, the output values for the variables $M$, $D_t$ and $\beta$, shown in italics.

| Run | $v$ (km/h) | $M$ ($\phi$) | $D_t$ (m) | $\beta$ ($\phi$) | Run | $v$ (km/h) | $M$ ($\phi$) | $D_t$ (m) | $\beta$ ($\phi$) |
|---|---|---|---|---|---|---|---|---|---|
| 1 | 500 | 0.40 | 6.05 | $3.65 \times 10^{-1}$ | 11 | 880 | 0.71 | 13.02 | $7.90 \times 10^{-2}$ |
| 2 | 538 | 0.44 | 6.40 | $3.27 \times 10^{-1}$ | 12 | 918 | 0.74 | 14.66 | $6.23 \times 10^{-2}$ |
| 3 | 576 | 0.47 | 6.78 | $2.91 \times 10^{-1}$ | 13 | 956 | 0.77 | 16.76 | $4.76 \times 10^{-2}$ |
| 4 | 614 | 0.50 | 7.22 | $2.57 \times 10^{-1}$ | 14 | 994 | 0.80 | 19.52 | $3.51 \times 10^{-2}$ |
| 5 | 652 | 0.53 | 7.71 | $2.25 \times 10^{-1}$ | 15 | 1032 | 0.84 | 23.33 | $2.46 \times 10^{-2}$ |
| 6 | 690 | 0.56 | 8.28 | $1.95 \times 10^{-1}$ | 16 | 1070 | 0.87 | 28.89 | $1.60 \times 10^{-2}$ |
| 7 | 728 | 0.59 | 8.94 | $1.68 \times 10^{-1}$ | 17 | 1108 | 0.90 | 37.78 | $9.37 \times 10^{-3}$ |
| 8 | 766 | 0.62 | 9.70 | $1.42 \times 10^{-1}$ | 18 | 1146 | 0.93 | 54.24 | $4.55 \times 10^{-3}$ |
| 9 | 804 | 0.65 | 10.61 | $1.19 \times 10^{-1}$ | 19 | 1184 | 0.96 | 95.00 | $1.48 \times 10^{-3}$ |
| 10 | 842 | 0.68 | 11.69 | $9.78 \times 10^{-2}$ | 20 | 1222 | 0.99 | 364.88 | $1.01 \times 10^{-4}$ |

The following table is a variation of Table A3. Here, the $L_t$ column has substituted the v column and there are five fewer runs because $L_t$ adopts three values for each number of containers (15 rows in total):

**Table A5.** Input columns, similar to those of Table A3 and output columns. The output variables are shown in italics: $E'_t$ and $I_E$ are for reference and $e'_t$ and $I_C$ serve to elaborate the curves.

| Run | $n_{cont}$ ($\phi$) | $f_1$ ($\phi$) | $f_2$ ($\phi$) | $f_3$ ($\phi$) | $f_4$ ($\phi$) | $f_5$ ($\phi$) | $L_t$ (km) | $m_{Li^+}$ (kg) | $\mathbf{m_{EB}}$ (kg) | $E'_t$ (kWh/km) | $I_E$ (kWh/t) | $e'_t$ (kWh/tkm) | $I_C$ (t/h) | $I_C^{-1}$ (h/t) |
|---|---|---|---|---|---|---|---|---|---|---|---|---|---|---|
| 1 | 1 | 1 | 0 | 0 | 0 | 0 | 500 | 350 | 750 | 1.07 | 18.86 | $3.77 \times 10^{-2}$ | 42.20 | $2.37 \times 10^{-2}$ |
| 2 | 1 | 1 | 0 | 0 | 0 | 0 | 750 | 350 | 750 | 0.93 | 24.74 | $3.30 \times 10^{-2}$ | 28.19 | $3.55 \times 10^{-2}$ |
| 3 | 1 | 1 | 0 | 0 | 0 | 0 | 1000 | 350 | 750 | 0.87 | 30.62 | $3.06 \times 10^{-2}$ | 21.16 | $4.73 \times 10^{-2}$ |
| 4 | 2 | 1 | 1 | 0 | 0 | 0 | 500 | 400 | 1000 | 1.73 | 15.28 | $3.06 \times 10^{-2}$ | 84.40 | $1.18 \times 10^{-2}$ |
| 5 | 2 | 1 | 1 | 0 | 0 | 0 | 750 | 400 | 1000 | 1.50 | 19.90 | $2.65 \times 10^{-2}$ | 56.38 | $1.77 \times 10^{-2}$ |
| 6 | 2 | 1 | 1 | 0 | 0 | 0 | 1000 | 400 | 1000 | 1.39 | 24.53 | $2.45 \times 10^{-2}$ | 42.33 | $2.36 \times 10^{-2}$ |
| 7 | 3 | 1 | 1 | 1 | 0 | 0 | 500 | 450 | 1250 | 2.39 | 14.08 | $2.82 \times 10^{-2}$ | 126.60 | $7.90 \times 10^{-3}$ |
| 8 | 3 | 1 | 1 | 1 | 0 | 0 | 750 | 450 | 1250 | 2.07 | 18.29 | $2.44 \times 10^{-2}$ | 84.57 | $1.18 \times 10^{-2}$ |
| 9 | 3 | 1 | 1 | 1 | 0 | 0 | 1000 | 450 | 1250 | 1.91 | 22.50 | $2.25 \times 10^{-2}$ | 63.49 | $1.58 \times 10^{-2}$ |
| 10 | 4 | 1 | 1 | 1 | 1 | 0 | 500 | 500 | 1500 | 3.05 | 13.49 | $2.70 \times 10^{-2}$ | 168.80 | $5.92 \times 10^{-3}$ |
| 11 | 4 | 1 | 1 | 1 | 1 | 0 | 750 | 500 | 1500 | 2.64 | 17.48 | $2.33 \times 10^{-2}$ | 112.76 | $8.87 \times 10^{-3}$ |
| 12 | 4 | 1 | 1 | 1 | 1 | 0 | 1000 | 500 | 1500 | 2.43 | 21.48 | $2.15 \times 10^{-2}$ | 84.65 | $1.18 \times 10^{-2}$ |
| 13 | 5 | 1 | 1 | 1 | 1 | 1 | 500 | 550 | 1750 | 3.72 | 13.13 | $2.63 \times 10^{-2}$ | 211.01 | $4.74 \times 10^{-3}$ |
| 14 | 5 | 1 | 1 | 1 | 1 | 1 | 750 | 550 | 1750 | 3.21 | 17.00 | $2.27 \times 10^{-2}$ | 140.95 | $7.09 \times 10^{-3}$ |
| 15 | 5 | 1 | 1 | 1 | 1 | 1 | 1000 | 550 | 1750 | 2.95 | 20.87 | $2.09 \times 10^{-2}$ | 105.81 | $9.45 \times 10^{-3}$ |

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
