# Peer review of "Analysis of the Effectiveness of a Freight Transport Vehicle at High Speed in a Vacuum Tube (Hyperloop Transport System)"

_algorithms, doi:10.3390/a17010017_

Round 1
Reviewer 1 Report
Comments and Suggestions for Authors
This paper presents an analysis of the effectiveness of a freight transport vehicle at high speed in a vacuum tube (hyperloop transport system).
The article was written to a high standard. The article meets the objective of scientificity.
Author Response
Thank you for your positive review. We have further improved the article to meet the requirements set by the other reviewers.

Reviewer 2 Report
Comments and Suggestions for Authors
The current paper discuss a numerical model development to be used for cargo transport capacity of vehicle operated at high speed. There are some vital shortcoming in the paper must be addressed first as follows:
1. Page 2, lines 51-62, add word of Ref before citation number at the begining.
2. The literature review needs to be extended to cover similar research at the last 5 years with focusing on the gaps which found.
3. Lines 88-92, the sentence has a grammer error, check and correct it.
4. Why the first citation of figure 1 appeare in page 8 while the figure itself is in page 4? Correct it.
5. Add references for equations presented in the article.
6. Repeat the numbering of equations to be rearranged in order as appeared in the paper, give another numerical numbers for equations in the appendix, for example 1A.
7. Discuss the results presented in Table 3.
8. Add more graphs to discuss the results in a scentific way not just numbers presentation.
9. Discuss the ability to extend the current work for other types of vehicles with emphasing on the difference between them.
Author Response
Thank you for identifying the vital shortcomings that our article had. Your feedback was very valuable to us and really helped us to improve the article. In the attached file we reply to your particular comments:

Reviewer 3 Report
Comments and Suggestions for Authors
Please see the attached file.

Comments on the Quality of English Language
Corrections are needed.
Author Response
Thank you for identifying the main flaws that our article had. Your request for corrections was very valuable to us and really helped us to improve the article. In the attached file we reply to your particular comments.

Round 2
Reviewer 3 Report
Comments and Suggestions for Authors
The authors carefully followed the requested comments. The paper is recommended for publication.
Comments on the Quality of English Language
fine